# Vertebrate blood cell volume increases with temperature: implications for aerobic activity

James F. Gillooly and Rosana Zenil-Ferguson

Department of Biology, University of Florida, Gainesville, FL, USA

## ABSTRACT

Aerobic activity levels increase with body temperature across vertebrates. Differences in these levels, from highly active to sedentary, are reflected in their ecology and behavior. Yet, the changes in the cardiovascular system that allow for greater oxygen supply at higher temperatures, and thus greater aerobic activity, remain unclear. Here we show that the total volume of red blood cells in the body increases exponentially with temperature across vertebrates, after controlling for effects of body size and taxonomy. These changes are accompanied by increases in relative heart mass, an indicator of aerobic activity. The results point to one way vertebrates may increase oxygen supply to meet the demands of greater activity at higher temperatures.

## INTRODUCTION

The tremendous variation in aerobic activity levels among vertebrates is related to differences in body temperature (*Bennett & Ruben, 1979*; *Bennett, 1987*). Colder-bodied species tend to be more sluggish or sedentary, whereas warmer-bodied species tend to show higher levels of aerobic activity. From an ecological perspective, differences in levels of aerobic activity underlie many differences in species' lifestyles, including feeding modes, movement patterns and rates of locomotion (*Bennett, 1980*; *Filho et al., 1992*; *Angilletta Jr, Huey & Frazier, 2010*; *Hein, Hou & Gillooly, 2012*). From an evolutionary perspective, these differences may lead to greater fitness (*Kingsolver & Huey, 2008*; *Angilletta Jr, Huey & Frazier, 2010*). For this reason, the greater aerobic activity afforded by higher temperatures is often cited as an explanation for the evolution of endothermy (*Bennett & Ruben, 1979*; *Clarke & Portner, 2010*). Yet, we have much to learn about how temperature may influence the structure or function of cardiovascular systems, and how this in turn may affect aerobic activity levels (*Gledhill, Warburton & Jamnik, 1999*; *Clarke & Portner, 2010*; *Hillman, Hancock & Hedrick, 2013*).

High levels of aerobic activity are generally thought to be limited by oxygen delivery (*Di Prampero, 1985*; *Farrell, 2002*), which is a function of arterial oxygen content and cardiac output (*Gledhill, Warburton & Jamnik, 1999*). Still, it remains unclear how temperature affects delivery. Arterial oxygen content is largely determined by hematocrit, which

Corresponding author
James F. Gillooly, gillooly@ufl.edu

for most vertebrates falls with the fairly narrow range of 30–45% (*Gledhill, Warburton & Jamnik, 1999*; *Stark & Schuster, 2012*), but how it may vary with temperature or activity level remains unclear (*Gledhill, Warburton & Jamnik, 1999*; *Schmidt & Prommer, 2008*; *Stark & Schuster, 2012*). Cardiac output is a function of both heart rate and stroke volume, but it is known to increase with both temperature and activity level (*Bishop, 1997*; *Farrell, 1991*; *Vinogradov & Anatskaya, 2006*; *Hillman, Hancock & Hedrick, 2013*). Temperature is thought to effect cardiac output mainly through its effect on heart rate, whereas blood volume is thought to influence stroke volume (*Gledhill, Warburton & Jamnik, 1999*). Yet, while blood volume is considered to be greater in species with higher aerobic activity levels (*Hillman, 1976*; *Prothero, 1980*; *Filho et al., 1992*; *Gallaugher & Farrell, 1998*; *Dawson, 2005*), and higher activity levels are associated with higher temperatures (*Bennett & Ruben, 1979*; *Bennett, 1987*), there is no known relationship of blood volume with temperature among vertebrates.

Previous work in mammals has shown that maximum rates of oxygen consumption, a measure of maximum aerobic activity, are highly correlated with the total volume of red blood cells in the body (*Birchard, 1997*). Since most oxygen-carrying hemoglobin is found in red blood cells, and the cellular density of hemoglobin varies relatively little across species with different-sized cells (*Hawkey et al., 1991*), this would imply more aerobic species have relatively greater body hemoglobin content as suggested by *Birchard (1997)*. Furthermore, if one assumes that hematocrit is relatively constant in mammals, this would imply that more aerobic mammals have greater total blood volumes. This result by *Birchard (1997)* suggests the possibility of a relationship between red blood cell volume and temperature given the association between temperature and activity level. If such a relationship exists, this could help explain why *RRBC* varies by more than an order of magnitude across vertebrates (*Bond & Gilbert, 1958*; *Thorson, 1961*; *Thorson, 1968*).

Here we examine if the percentage of body mass comprised of red blood cells (i.e., *RRBC* in % body mass) increases with body temperature across species. In the case of ectotherms, of course, body temperature reflects the environmental temperatures in which they live. In addition, we examine if/how differences in *RRBC* may be related to differences in relative heart mass (*RHM*; in % body mass). Heart mass has been shown to be linearly related to stroke volume in mammals (*Holt, Rhode & Kines, 1968*). And relative heart mass, like *RRBC*, tends to be greater in more aerobic species (*Farrell, 1991*; *Bishop, 1999*; *Vinogradov & Anatskaya, 2006*). In performing both analyses, we consider the possible effects of body mass and taxonomy. Effects of body mass are a particularly important consideration since mass-specific blood volume is often considered to be independent of body mass (*Prothero, 1980*; *Dawson, 2005*), whereas mass-specific rates of maximum oxygen consumption are often considered to decrease weakly with increasing mass (*Bishop, 1999*; *Weibel et al., 2004*). Our hope is that these analyses, which include data from all major vertebrates groups (fishes, amphibians, reptiles, birds, and mammals), will provide a step toward better understanding the observed variation in red blood cell volume among vertebrates.

## METHODS

### Data

Data on *RRBC* were compiled from published sources for vertebrates from a range of habitat types (freshwater, marine, terrestrial), and span a broad range of body sizes, temperatures, taxonomic affiliations, and aerobic activity levels (Appendix S1). Data were originally collected using indicator dilution or labeling methods (*Zierler, 2000*), and originally expressed as either percent body mass or ml RBC/100 g body mass. These units are assumed to be equivalent in our analyses since the specific gravity of red blood cells is only slightly greater than 1 (*Trudnowski & Rico, 1974*). Analyses were restricted to adult or sub-adult individuals as values may change through early ontogeny (*Garland & Else, 1987*). Analyses also did not include (i) air-breathing divers (e.g., marine mammals, sea turtles) given their exceptional nature (*Costa, Gales & Crocker, 1998*), (ii) elasmobranchs given the confounding effects of low albumin concentration on measurement (*Tort et al., 1991*), (iii) Antarctic fish with little or no hemoglobin, and (iv) urodeles given the large fraction of blood cells held in the spleen (see Discussion) and the difficulty in obtaining estimates of preferred body temperature. To the best of our knowledge, all other available data on *RRBC* were included in the analyses for which body size and temperature estimates were also available.

Estimates of relative heart mass were taken from studies on adult individuals of the same species. On occasion, heart mass was estimated from ventricular mass assuming that ventricle mass comprised 60% of heart mass (*Santer & Walker, 1980*; *Seymour, 1987*; *Brill & Bushnell, 1991*) (see Appendix S1).

Body mass estimates from the original studies of *RRBC* were used. For endotherms (birds and mammals), resting body temperatures were used (primary source: *Clarke & Rothery, 2008*), and assumed to be roughly equivalent to active body temperatures. For ectotherms (amphibians, reptiles, and fishes), the preferred environmental or body temperatures were used assuming these are the temperatures at which species are typically active. If temperature estimates for a particular species were unavailable, the temperature of one or more species from the same genus was used based on the mean value. Finally, in the case of fishes, the temperature at which a species was held was sometimes used as the preferred temperature (see Appendix S1).

### Analyses

To evaluate the body size and temperature dependence of *RRBC*, and the dependence of *RRBC* on relative heart mass, we first performed type II nested ANOVAS (*Sokal & Rohlf, 1969*) to account for possible effects of evolutionary relatedness among species. In the absence of a well-resolved vertebrate phylogeny, nested ANOVAS are the preferred method of analysis to address this issue (*Harvey & Pagel, 1991*).

With the nested ANOVAS, we examined the influence of taxonomic class, order within class, and family within order, to determine the appropriate taxonomic level for analysis for the variables in question. This analysis revealed that significant variation in three of the four variables could be explained at the level of taxonomic order (Table 1). Thus,

**Table 1 Statistics describing the body mass and temperature dependence of relative red blood cell volume (i.e., *RRBC*).** Models were fit using weighted least squares regression at the level of taxonomic order based on the results from Type II Nested ANOVA (**a**, see Methods). Models were expressed as: $\ln(RRBC) = a\ln(M) + bT + C$, where *RRBC* represents relative red blood cell volume (% body mass), *M* represents body mass (g), and *T* represents temperature (°C). The coefficients *a*, *b* and *C* from each model are listed below for **b** all species (i.e., ectotherms and endotherms), **c** for ectotherms only, and **d** for endotherms only. Data are provided in Appendix S1.

**a. TYPE II NESTED ANOVA**

| Variables | F, at order level | p-value |
|---|---|---|
| Temperature | 3.363 | 0.011 |
| $\ln(M)$ | 10.718 | $1.95 \times 10^{-5}$ |
| $\ln(RRBC)$ | 2.080 | 0.079 |
| $(RHM)$ | 10.686 | 0.037 |

**b. ALL SPECIES**

| Coefficient | Estimate | Std. Error | T-statistic | p-value |
|---|---|---|---|---|
| $C$ | −0.959 | 0.209 | −4.597 | $1.06 \times 10^{-4}$ |
| $a$ | −0.062 | 0.018 | −3.364 | 0.002 |
| $b$ | 0.065 | 0.005 | 11.838 | $9.6 \times 10^{-12}$ |
| R-squared 0.85 | | | | |
| F-statistic 73.11 | | D.F: 2, 25 | | p-value: $3.58 \times 10^{-11}$ |

**c. ECTOTHERMS**

| Coefficient | Estimate | Std. Error | T-statistic | p-value |
|---|---|---|---|---|
| $C$ | 0.209 | 0.240 | 0.869 | 0.407 |
| $a$ | −0.153 | 0.022 | −6.793 | $7.97 \times 10^{-5}$ |
| $b$ | 0.041 | 0.006 | 7.211 | $5.03 \times 10^{-5}$ |
| R-squared | 0.94 | | | |
| F-statistic | 70.48 | D.F: 2, 9 | | p-value: $3.17 \times 10^{-6}$ |

**d. ENDOTHERMS**

| Coefficient | Estimate | Std. Error | T-statistic | p-value |
|---|---|---|---|---|
| $C$ | −1.049 | 1.315 | −0.798 | 0.439 |
| $a$ | −0.049 | 0.022 | −2.224 | 0.044 |
| $b$ | 0.066 | 0.032 | 2.051 | 0.061 |
| R-squared | 0.49 | | | |
| F-statistic | 6.26 | D.F: 2,13 | | p-value: 0.012 |

for both analyses, we fit weighted linear regressions using mean values at the level of taxonomic order. The regression is weighted depending on the proportion of taxa within each order. For the relationship between size, temperature and *RRBC*, we fit a model of the form: $\ln RRBC = a\ln M + bT + C$. Here *a* is a body-mass scaling exponent, *b* (°C$^{-1}$) characterizes the exponential temperature dependence, and *C* is a taxon-specific constant that includes random error. The variables *M* (g) and *T* (°C) in this formulation are mean values of body mass (g) and temperature (°C) at the order level. Similarly, to describe the relationship between *RRBC* and relative heart mass, we fit a model of the

form: $\ln(RRBC) = a\ln(RHM) + C$. To statistically evaluate this model, we also used weighted least squares regression on data at the order level. For both linear models, to further investigate the effects of taxonomic order on our results, we performed a bootstrap analysis. The bootstrap analysis consisted of resampling taxa within each order and recalculating the mean values for each of the variables in the model (*Efron & Tibshirani, 1993*). Weighted linear regressions were then estimated with these new values after 30,000 repetitions produced consistent estimates.

To graphically represent the effects of body size and temperature on *RRBC*, we divided *RRBC* through by the observed body mass-dependence, and then plotted the natural logarithm of this new "body-mass corrected" value against temperature (i.e., $y = \ln[RRBC/M^a]$). In the plots, we present all species-level data, but lines are only fitted to the data at the order level.

## RESULTS

*RRBC* varied with both body mass (*M*, in grams) and temperature (*T*, in °C) across the 65 species considered here (birds: $n = 9$; mammals: $n = 16$; reptiles: $n = 14$; amphibians: $n = 6$; fishes: $n = 20$) following the equation: $\ln(RRBC) = -0.062\ln(M) + 0.065T - 0.96$. Together, the two variables explained 85.4% of the variation in *RRBC* (range: 0.6–6.9% body mass) based on weighted multiple regression of ln-transformed data at the level of order (see Table S1). Both showed statistically significant, independent effects on *RRBC*. As indicated by the equation, *RRBC* decreased with body mass, $M(g)$, as $RRBC \propto M^{-0.062}$, and increased exponentially with temperature, $T$ (°C), as $RRBC \propto e^{0.065T}$ across all species. Figure 1 shows a plot of the natural logarithm of body-mass-corrected relative red blood cell volume (i.e., $\ln[RRBC/M^{-0.062}]$) versus temperature. The fitted line ($y = 0.065T - 0.96$) indicates that, after accounting for the effects of body mass, *RRBC* increases by about an order of magnitude from 5 to 40 °C. When endotherms and ectotherms were analyzed separately, their body mass and temperature dependencies were somewhat different than was observed across all species (Table 1). Temperature and body mass showed significant, independent effects on *RRBC* among ectotherms, but temperature was not a significant predictor among endotherms ($p = 0.06$; Table 1).

*RRBC* was also significantly, positively correlated with relative heart mass (Fig. 2; $F = 51.99$, 1 and 22 D. F.; $p = 3.16 \times 10^{-7}$) among the 33 species considered in this analysis (birds: $n = 8$; mammals: $n = 11$; reptiles: $n = 6$; amphibians (anurans): $n = 4$; bony fishes: $n = 5$). The increase in *RRBC* with relative heart mass was less than proportional as indicated by the slope being <1. ($RRBC = 0.73\,RHM + 1.43$; $r^2 = 70.2\%$). Bootstrap analyses for the relationships shown in both Figs. 1 and 2 yielded very similar relationships (Fig. 1: $y = 0.065T - 0.95$; Fig. 2: $y = 0.71\,RHM + 1.45$).

## DISCUSSION

The results shown in Fig. 1 indicate that much of the variation in *RRBC* among vertebrates can be explained by differences in body size and temperature. They may also provide a first step toward better understanding how vertebrates meet the oxygen demands of greater activity at higher temperatures. At present, we lack a clear understanding of how

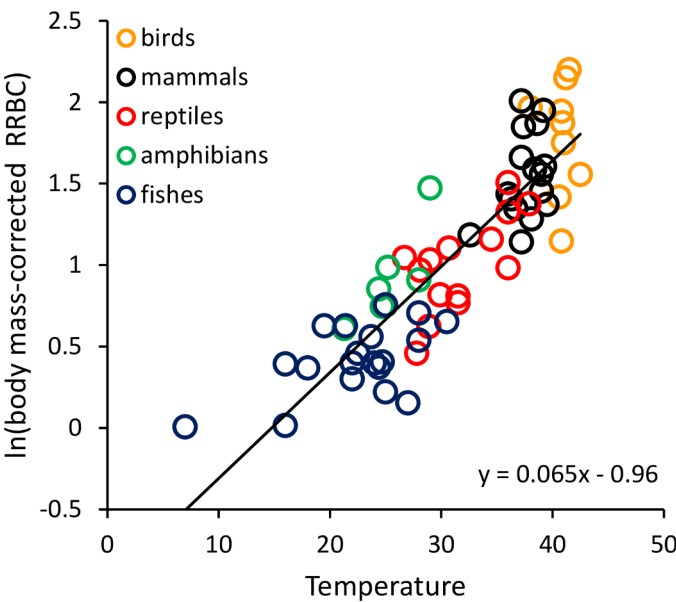

**Figure 1 The natural logarithm of body mass-corrected relative red blood cell mass (*RRBC*) vs. temperature in vertebrates.** The regression line shown (i.e., $\ln(RRBC/M^{-0.06}) = 0.065T - 0.96$) is based on weighted values for data averaged at the level of taxonomic order for the 65 species shown here. Relative red blood cell volume (*RRBC*) is expressed as a percentage of body mass, temperature (*T*) is expressed in degrees Celsius, and body mass (*M*) is expressed in grams.

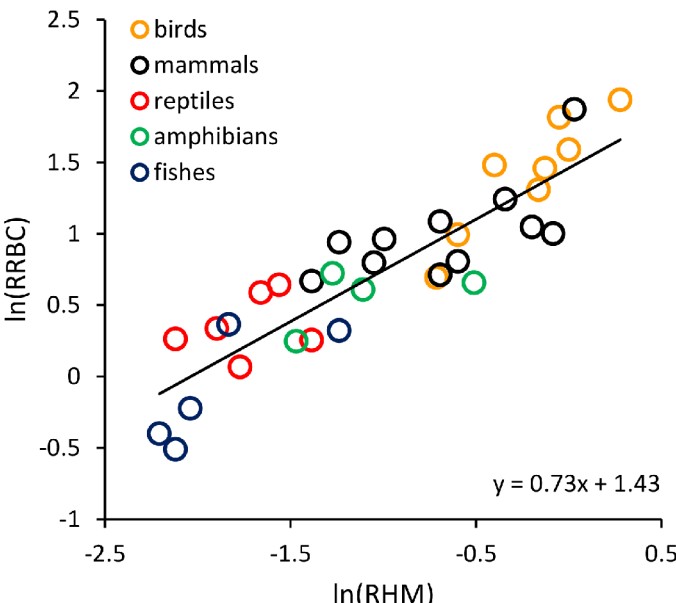

**Figure 2 The natural logarithm of relative red blood cell volume (*RRBC*; % body mass) vs. the natural logarithm of relative heart mass (*RHM*; % body mass) in vertebrates.** The regression line shown (i.e., $\ln(RRBC) = 0.73\,RHM + 1.43$) is based on weighted values for data averaged at the level of taxonomic order.

stroke volume and heart rate combine to determine maximum levels of cardiac output, and constrain aerobic activity, across temperatures. These results suggest that, in addition to increasing heart rates at higher temperatures, vertebrates may also increase the total volume of red blood cells to augment oxygen delivery. As previously pointed out by *Birchard (1997)*, these increases in *RRBC* are consistent with increases in stroke volume. This is because blood volume is known to influence stroke volume, and *RRBC* constitutes a relatively constant fraction of blood volume in most vertebrates (i.e., 30–45%). The relationship between relative heart mass and relative red blood cell volume shown in Fig. 2 is also consistent with a relationship between *RRBC* and stroke volume.

However, there is significant variation about the fitted lines in Figs. 1 and 2, and clear differences among taxonomic groups. These may reflect the variety of factors that may contribute to differences in *RRBC* among vertebrates. For example, to some degree, *RRBC* is phenotypically plastic. Species including our own may increase blood volume with exercise (*Lillywhite & Smits, 1984*; *Convertino, 1991*; *Schmidt & Prommer, 2008*). Species may also adjust *RRBC* in response to other environmental challenges, such as hypoxia. In urodeles, for example, a large fraction of red blood cells may be held in the spleen and only released at high temperatures for just this reason (*Tort et al., 1991*). Thus, in amphibians, desiccation resistance may tradeoff with aerobic activity (*Hillman, Withers & Drewes, 2000*).

Future work on this topic that more directly compares *RRBC* with mass-specific maximum oxygen consumption rates would benefit our understanding of the effects of temperature on cardiac output and aerobic activity. Research that considers the body size and temperature dependence of maximum stroke volume, maximum heart rate, and maximum oxygen consumption rate, across diverse species is also needed. Here we observed an increase in *RRBC* with temperature (i.e., $Q_{10}$ of approx. 1.9) that is less than the temperature-dependence of oxygen consumption described for resting vertebrates (i.e., $Q_{10}$ of approx. 2.7 by *Gillooly et al., 2001*). But, the temperature dependence of maximum oxygen consumption is not well-established. Thus, this difference may indicate that the temperature-dependence of oxygen consumption is weaker at high activity levels, or alternatively, that other changes in the structure or function of cardiovascular system increase the temperature-dependence of oxygen supply. In the latter case, temperature effects on heart rate would perhaps be the most likely culprit, but many factors that affect oxygen supply rate could change with temperature (e.g., blood viscosity, oxygen dissociation; *Sidell, 1998*).

Still, the observed body size and temperature dependence of *RRBC* raise questions regarding possible relationships between temperature, cardiovascular design, and aerobic activity level. To some degree, our current understanding of cardiovascular design is based on models that do not consider temperature (e.g., *Spatz, 1991*; *Dawson, 2005*). Temperature is generally considered to affect the dynamics of cardiovascular systems through its effects on biochemical kinetics and associates rates (e.g., heart rate)—not the volume or mass of the component parts of these systems (*Kingsolver & Huey, 2008*). Yet, the body mass dependence of *RRBC* shown here is similar to that reported for maximum oxygen consumption in some groups (*Bishop, 1999*; *Weibel et al., 2004*), and the

temperature dependence is at least close to that which is typically assumed for resting or maximum oxygen rates (i.e., $Q_{10}$ of 2–3). We thus speculate that perhaps natural selection acts to alter the design of some features of cardiovascular systems across gradients in temperature to maximize oxygen supply rates and allow for higher aerobic activity.

## ACKNOWLEDGEMENTS

We thank Andrew Clarke for graciously sharing his data on endotherm body temperatures, and Andrew Allen for helpful discussions.

### Funding

No funding sources were used for this work.

### Competing Interests

The authors declare there are no competing interests.

### Author Contributions

- James F. Gillooly conceived and designed the experiments, performed the experiments, analyzed the data, wrote the paper, prepared figures and/or tables, reviewed drafts of the paper.
- Rosana Zenil-Ferguson performed the experiments, analyzed the data, wrote the paper, prepared figures and/or tables, reviewed drafts of the paper.

### Supplemental Information

Supplemental information for this article can be found online at http://dx.doi.org/10.7717/peerj.346.

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
