# Peer review of "Vertebrate blood cell volume increases with temperature: implications for aerobic activity"

_PeerJ, doi:10.7717/peerj.346_

## Round 0.1 · original submission · Major Revisions

Like the reviewers, I find this study interesting and worth publishing, so thank you for submitting to PeerJ. However, in addition to addressing each of the reviewer's points in your revision (I especially agree that toning down implications of causation in some correlations is needed as per Rev 2), I ask that you do the following:

1) Present all of the data analyzed in the paper in tabular form as a supplementary online appendix, so others may use those data if needed. Ensure that if the data come from other studies, those studies get cited as a source in a column of that/those table(s). Where values are estimates, those should be noted in the table(s) and explained in the caption(s). Otherwise, as it stands the paper is not reproducible in a modern sense.

2) Explicitly address whether you required IACUC approval for any data collected originally for this study. it does not seem so, but a sentence stating that only literature data were used and not any novel measurements requiring such approval would help.

I consider #1 and #2 necessary standards in this field/in science at present, and they will substantially improve this already useful study and increase the likelihood it is cited frequently.

·

Basic reporting

Line 24: “species must overcome the weak temperature dependence of passive diffusion relative to oxygen consumption”. I don’t understand this – please take a little space to elaborate. Also please explicate the next sentence – how could it be that diffusion limit activity levels? This idea of limits being set by diffusion appears several times in the paper and deserves a bit more space and a clearer exposition.

Line 37: omit “also”

I found it a bit confusing to refer to omit ‘specific’ when referring to red blood cell volume. Could the acronym be SRBC, or RRBS if ‘relative’ is preferred to ‘specific’.

Personally I like to look at the data early on in the Results, so I would have preferred reference to and description of fig. 1 early on in the Results section.

The paragraph starting on line 109 is confusing. I think lines 109 – 114 are intended as discussion of the previous para. Perhaps they could be related to the previous para in a more logical way.

The paragraph starting lin line 118 reports the correlation of RBC with relative heart mass, but this has already been reported in the the previous paragraph and fig. 2. Perhaps everything to do with heart mass results could go together into a single para.

The para starting in line 123 says inter alia that previous work has shown that relative heart mass has the same depence on size and temperature as RBC, and refers to unpublished work to support this statement. Perhaps the words ‘previous work has reported that’ could be removed.

On balance I think I would prefer reporting all the results before starting to discuss them. That way the discussion could perhaps start by restating the three main results in a sentence or two (RBS increases exponentially with temperature and shows a small decrease with body mass, RBC increases with relative heart mass) so we all know what we are talking about. Then I think the current discussion ideas could be elaborated a little, they are a bit congested at presented. Also I was disappointed not to find a discussion of the results in the context of the Arrhenius function and activation energy which Gillooly pioneered in his earlier work.

Figure 2: Suggest transfer last sentence to the Results section.

Experimental design

No comments

Validity of the findings

No comments

Additional comments

I found this paper very interesting and thank the authors for their work

·

Basic reporting

The authors have a strong track record of influential papers. This one is generally professionally written and clear. It provides a novel study of red blood cell volume. However, I do not think that it is interpreted correctly, as stated below.

Experimental design

No comments.

Validity of the findings

General comments
This is an interesting analysis of the correlation between red blood cell volume, body mass, relative heart mass and temperature in vertebrates. It is the first that I know of to examine blood cell volume systematically. The analysis seems to be done correctly, although some would dispute that an adequate phylogenetic tree for the vertebrates to the order level is not available.
The major criticism I have is that the conclusion misleads me and probably other potential readers. I am not convinced that temperature directly determines red blood cell volume. This conclusion appears throughout the presentation, from the title, the abstract and much of the discussion. The introduction implies that body temperature is a major factor in determining aerobic metabolic scope (oddly called ‘athleticism’ here). However, mitochondrial surface area is the major determinant: it is 4 times higher in mammals than in reptiles (Else, P. and Hulbert, A. (1985). An allometric comparison of the mitochondria of mammalian and reptilian tissues: The implications for the evolution of endothermy. Journal of Comparative Physiology B: Biochemical, Systemic, and Environmental Physiology 156, 3-11.), for example, but the effect of temperature is only about 2-fold over a temperature range of less than about 10 K for these groups (Fig 1).
More importantly, the study shows only correlations, not functional relationships. For example, one could plot RBC volume on arterial blood pressure in vertebrates and get a similar correlation, but it would have no causal connection. Or, plot RBC on heart mass as this paper does (Fig. 2) and try to imply causation. You can’t. Many things increase in animals with higher capacity for aerobic work, but none of them is directly influenced by temperature, except the rate of biochemical reactions.
The abstract says: “Here we show that vertebrates provide more oxygen to tissues at higher temperatures in part by increasing the total volume of red blood cells in the body.” This needs to be clarified here and in the bulk of the paper, because, as stated, it is wrong. I think the point is that animals with higher aerobic scope have higher oxygen capacitance (=carrying capacity) of the blood. Hematocrits may be higher. Hemoglobin concentrations may be higher. It should be made clear how the increase in RBC volume is attained, either by increasing the hematocrit or by increasing total blood volume. It is important, because if total red blood cell volume is increased simply by increasing total blood volume, then it would not augment oxygen delivery. The study should also consider the fact that mean cellular hemoglobin (MCH) concentration differs between animals. I suspect that Antarctic fish were removed from the dataset because of low MCH. The point I make is that it is not the volume of oxygen held in the total circulation that is important, it is the rate at which the oxygen is pumped around, and a higher blood oxygen carrying capacity is important, not total red blood cell volume.
The behavior of the animal in its ecological environment determines energy and oxygen demands of activity, and the cardiorespiratory system has evolved to supply it with the least energy cost. Aerobic energy production is primarily related to mitochondrial volume which must be supplied with oxygen by the structure of the oxygen cascade, from the gas exchanger, through the circulatory system and finally diffusion through the tissue to the mitochondria. The barriers of the oxygen cascade have been analysed in mammals by Weibel et al. and the present authors point out that the resistance to oxygen flow at each level might be matched. So why do they insist on saying that their variables are influenced by temperature alone?

Additional comments

Specific comments
Lines
32-33 This statement should refer to mass-specific RBC volume, I think.
59 If the point of the paper concerns maximum metabolic rate, shouldn’t the active body temperature be used?
98 Values expressed on a 100 g, mass-specific basis is somewhat old-fashioned and not close to SI recommendations. It should be on a gram or kilogram basis, if anything. Of course, mass-specific values do not eliminate the effect of body mass, so I would do the analysis on whole-animal RBC volumes.
101 What are the units for these numbers? Also here it says ‘log-transformed’, not ‘ln-transformed’. Which is it? It is not clear why natural logs are used and not logs to the base 10. The latter produces graphs with ticks that have convenient meaning. For example, the log of 100 kg is 2, and it is easy to find 100 kg on a graph. The natural log of 100 kg is 4.60517.
103 Please provide the entire allometric equation, rather than just the proportional sign. That would enable someone to actually use the equation. It would be good to present the raw data for RBC on body mass in a figure, preferably without mass-specific units.
109-114 The first sentence says that the rate of oxygen supply increases at higher activity levels, but the data are for RBC volume, not a rate. The second sentence considers heart rate, which is absent altogether. Where in the paper does it show that smaller bodied species have higher RBC volume?
115-117 There should be a better foundation for this speculation. The heart size in Figure 2 is probably more related to higher blood flow rates and blood pressures than to viscosity directly.
131-132 I do not understand this sentence.
145 It is not clear to me how this study relates at all to the question of diffusion limitation.

---

## Round 0.2 · Minor Revisions

Thanks for the speedy revision. 1 reviewer is satisfied; the other is even less convinced, it seems. I am calling my decision "minor revisions" but I think you should revise to do a better job of addressing reviewer Seymour's points-- there may just have to be a lingering disagreement but his points do deserve to be incorporated in the paper where feasible. I think their points in general are quite constructive and deserve a second look. Then I will make my final decision once the revision is done.

·

Basic reporting

The function of the second paragraph in the paper (lines 19-26) is to make the case for the proposed analyses. I stumbled a bit over lines 21-26, which read:

“At moderate levels of activity, temperature effects on biochemical kinetics and related dynamics (e.g., increasing heart rate) allows species the ability to both consume more oxygen and supply more oxygen, and thus sustain greater activity (Gillooly et al., 2001; Kingsolver & Huey, 2008). However, more prolonged or intense activity is generally considered to be limited by oxygen supply (di Prampero, 1985). Temperature effects on biochemical kinetics may not be sufficient to meet high oxygen demands (Wagner et al., 1990; Farrell, 2002; Gjedde, 2010).”

I think the second sentence is saying that under prolonged or intense activity, temperature effects can no longer sustain the oxygen increase needed to fuel increased activity at higher temperatures. I think the third sentence elaborates on this. Please could the authors clarify, and so strengthen the case for the analyses described in the third and fourth paragraphs.

Experimental design

No comments

Validity of the findings

No comments

·

Basic reporting

The submission adheres to the policies. Since reviewing this paper I have reviewed another one by the same authors. It contained data on relative heart mass, apparently the same data as presented here. I suggest that the studies be combined, because the gist of the two papers is essentially identical.

Experimental design

No experiments were performed.

Validity of the findings

The data seem well collected and analysed.

Additional comments

Here are my responses to the author's rebuttal. I have made other annotations on the PDF of the new MS as well.
Overall, the new MS does not answer the two main criticisms of the first review: (1) I proposed that the structure and function of the circulatory system is primarily subservient to the requirement of the animal for aerobic activity, and is not directly related to temperature. (2) the argument that total red blood cell volume is related to the rate of oxygen delivery is flawed, because the volume of blood is not related to the rate that it flows. The authors do not address these adequately or correctly.
I may have gotten the ratio of mitochondria wrong between mammals and reptiles from this paper. I do not have the paper at hand to check. But this is not the main problem.
If, by the correlation between RBC volume and temperature, the paper concludes that temperature constrains athleticism, then the authors must also say that temperature affects arterial blood pressure, cardiac output, maximum aerobic scope, muscle myoglobin concentration, pulmonary surface area, diffusing capacity, a-v oxygen content difference, oxygen capacitance of the blood, tissue diffusive conductance, capillary density, etc., i.e. everything that relates to maximum aerobic capacity. Unfortunately they have not shown how total red blood cell volume relates functionally to oxygen delivery or aerobic capacity.
In fact, there is nothing in the paper about how temperature would influence total red cell volume. RBC volume does not augment the rate of oxygen delivery. Correlations are simply correlations unless one has a functional connection as to how one variable affects another.
Contrary to the author’s assurance, there is nothing in the 2nd paragraph of the introduction that shows how total red blood cell volume is relevant to oxygen delivery. There is an allusion to RBC volume to stroke volume in the discussion, but it does not hold water. There is no functional connection between blood volume and stroke volume. My attempts to explain that the oxygen capacitance of the blood IS relevant go unheeded.
The authors can choose not to accept my suggestions for improvement if they wish. For example, changing the values previously on a 100-g basis to percentages does not change the numbers. They justify this by stating that the specific gravity of blood is 1.0. This is wrong, as the paper they cite indicates.
I mentioned why the common log to the base 10 provides graphs that are easy (at least for me) to interpret. Logs can be made to any base and are equally simple to use. I know of no biological difference between them. Yet the authors say that natural logs are somehow more relevant to biology. This is strange. The suggestions I made were trying to make the paper more appreciated by physiologists like me. It is OK if the authors do not wish to take the advice.

---

## Round 0.3 · accepted · Accept

I am satisfied with the revised paper and see no case for further review- there are clearly substantial disagreements between the authors and one reviewer, but those can be sorted out in post-publication peer review and future literature. The paper has provided the key data that would be needed to test their results and conclusions, e.g. by using log10-based scaling. And I agree, it is clear that the authors have done a dilligent revision.